# *Francisella tularensis* in Wild Lagomorphs in Southern Spain’s Mediterranean Ecosystems

**DOI:** 10.3390/ani14233376

**Published:** 2024-11-23

**Authors:** Sabrina Castro-Scholten, Ignacio García-Bocanegra, Salvador Rejón-Segura, David Cano-Terriza, Débora Jiménez-Martín, Carlos Rouco, Leonor Camacho-Sillero, Antonio Arenas, Javier Caballero-Gómez

**Affiliations:** 1Grupo de Investigación en Sanidad Animal y Zoonosis (GISAZ), Departamento de Sanidad Animal, UIC Zoonosis y Enfermedades Emergentes ENZOEM, Universidad de Córdoba, 14014 Córdoba, Spain; sabrina1996cs@gmail.com (S.C.-S.); srejonsegura@gmail.com (S.R.-S.); davidcanovet@gmail.com (D.C.-T.); debora.djm@gmail.com (D.J.-M.); arenas@uco.es (A.A.); javiercaballero15@gmail.com (J.C.-G.); 2CIBERINFEC, ISCIII–CIBER de Enfermedades Infecciosas, Instituto de Salud Carlos III, 28029 Madrid, Spain; 3Departamento Biología Vegetal y Ecología, Área de Ecología, Universidad de Sevilla, 41012 Sevilla, Spain; c.rouco@gmail.com; 4Programa de Vigilancia Epidemiológica de la Fauna Silvestre en Andalucía (PVE), Consejería de Sostenibilidad, Medio Ambiente y Economía Azul, Junta de Andalucía, 29002 Málaga, Spain; leonorn.camacho@juntadeandalucia.es; 5Grupo de Virología Clínica y Zoonosis, Unidad de Enfermedades Infecciosas, Instituto Maimónides de Investigación Biomédica de Córdoba (IMIBIC), Hospital Universitario Reina Sofía, Universidad de Córdoba, 14004 Córdoba, Spain

**Keywords:** tularemia, vector-borne disease, wild lagomorphs, surveillance, zoonoses

## Abstract

*Francisella tularensis*, the causative agent of tularemia, is a zoonotic bacterium that is a concern for public and animal health. However, although wild lagomorphs are considered important reservoirs of this bacterium and are keystone species in the Iberian Peninsula, very little is known about the epidemiological role of these species. To address this research gap, a cross-sectional study was carried out, between the 2017/2018 and 2022/2023 hunting seasons, to investigate the occurrence of *F. tularensis* in spleen samples from 774 European wild rabbits (*Oryctolagus cuniculus*) and 178 Iberian hares (*Lepus granatensis*) inhabiting Iberian Mediterranean ecosystems. None of the 952 wild lagomorphs sampled showed the presence of *F. tularensis* DNA. Our results indicate low-to-no circulation of *F. tularensis* in European wild rabbit and Iberian hare populations, suggesting a limited risk of transmission to other sympatric species, including humans, in southern Spain.

## 1. Introduction

Tularemia, also known as rabbit or hare fever, is a multi-host vector-borne zoonoses caused by the highly virulent and infectious bacterium *Francisella tularensis*. The disease is often prolonged and debilitating, and it can be lethal to a wide variety of mammal species, including humans [1], who can acquire the infection through several routes: contact with infected animals, consumption of contaminated food or water, bites from infected vectors (such as ticks, flies, and mosquitoes), or inhalation of aerosols [2]. There are mainly two cycles of the disease that have been described: the terrestrial and aquatic cycles. In the terrestrial cycle, lagomorphs and rodents are the most important mammalian hosts, and different species of ticks and insects act as competent vectors. In Europe, the most common tick species involved belong to the genera *Dermacentor*, *Ixodes*, and *Haemaphysalis* [3]. Mosquitoes, especially those belonging to the genus *Aedes* [4]; flies; and, to a lesser extent, fleas have also been described as vectors of *F. tularensis* [5]. In the aquatic cycle, beavers, muskrats, and voles serve as hosts, contaminating the environment. Mosquitoes have been described as vectors connecting both cycles [6].

In Europe, the number of human *F. tularensis* cases has considerably increased during the last few years, with wildlife playing an important role in the epidemiology of the bacterium [7,8]. In fact, the European Centre for Disease Prevention and Control has recently emphasized the need for coordinated One Health surveillance to control tularemia in Europe, highlighting the importance of wildlife monitoring [8]. In this continent, zoonotic transmission from wild lagomorphs has already been reported, and contact with these species is recognized as one of the most significant risk factors for tularemia in humans [9,10,11,12]. Despite evidence emphasizing the role of wild lagomorphs as a zoonotic source of *F. tularensis* and their potential as sentinels for this bacterium [13], information about the role of these species in the epidemiology of tularemia is still scarce or unavailable in many European regions. In Spain, since the first tularemia outbreak was reported in 1997, which was related to the hunting and handling of hares [14], over 1000 human cases have been confirmed [15], with tularemia being currently listed as a notifiable disease and recognized as an emerging pathogen in this country [16]. Tularemia cases have been reported in wild lagomorphs in Spain, with almost all instances confined to the north–central regions of the country. Human outbreaks have already been linked to the presence of *F. tularensis* in these wild species in this country [17]. However, information regarding the circulation of this zoonotic bacterium in wild lagomorph species in southern Spain is very limited. Therefore, the aim of this study was to molecularly investigate the occurrence of *F. tularensis* in European wild rabbit (*Oryctolagus cuniculus*) and Iberian hare (*Lepus granatensis*) populations inhabiting Iberian Mediterranean ecosystems.

## 2. Materials and Methods

### 2.1. Study Area and Sampling Collection

Between the 2017/2018 and the 2022/2023 hunting seasons, a cross-sectional epidemiological study was conducted in Andalusia, southern Spain (87,300 km^2^, 36° N–38°60′ N, 1°75′ W–7°25′ W) (Figure 1). This region exhibits a Mediterranean climate characterized by hot and dry summers and mild winters. The western region presents higher mean humidity and less extreme mean temperatures than the central and eastern regions [18]. Andalusia is the Spanish region with the second highest number of wild lagomorphs hunted annually, with 1.4 million wild rabbits and 251 thousand hares harvested each year [19].

Sample size was calculated to ensure a 99% probability of detecting at least one positive animal, assuming a minimum prevalence of 0.5% in the study area [20]. We analyzed a total of 952 animals, including 774 European wild rabbits and 178 Iberian hares, from 135 hunting grounds. Samples were distributed across the eight provinces of Andalusia and provided by hunters from each of the hunting grounds during the study period. All animals were legally harvested during the hunting seasons (August–February). Spleens were removed from these animals aseptically and stored in individually labeled plastic tubes. The samples were kept refrigerated until arrival at the laboratory and immediately frozen at −80 °C until molecular analyses were undertaken.

During sampling, an epidemiological questionnaire was completed by directly interviewing the gamekeepers of the hunting grounds. Information on each animal was recorded, including species, location, year of sampling, age (determined by body weight and body length according to [21]), and sex.

### 2.2. Molecular Detection

Total DNA was extracted from spleen samples with the commercial NucleoSpin Tissue^®^ kit (Macherey-Nagel, Düren, Germany), following the manufacturer´s instructions. The concentration and quality of all DNA elutions were measured using the NanoDrop ND-2000 spectrophotometer (Thermo Fisher Scientific, Waltham, MA, USA). Values of 260/280 of all extractions were within the expected range, and the DNA presented a mean concentration of 407.7 ng/µL (Appendix A). The presence of *F. tularensis* DNA was tested by a broad-spectrum PCR using the Bio-Rad T100 Thermal Cycler (Hercules, CA, USA); the DreamTaq Green PCR master mix (2x) kit (Thermo Fisher Scientific^™^, Waltham, MA, USA); the primer set F5: 5′-CCT TTT TGA GTT TCG CTC C-3′ and F11: 5′-TAC CAG TTG GAA ACG ACT GT-3′, which targets 1140 bp within the 16S rRNA gene; and 8.0 µL of the extracted DNA [22]. The positive control was obtained after extracting the DNA of positive *Hyalomma lusitanicum* ticks already studied in [23] (GenBank Accession Number: MT386092). The PCR amplification conditions included initial denaturation at 95 °C for 2 min, followed by 35 cycles; each cycle consisted of denaturation at 95 °C for 30 s, annealing at 57 °C for 30 s, and extension at 72 °C for 1 min, with a final extension of 5 min at 72 °C. The PCR products were subjected to electrophoresis on a 1.5% agarose gel stained with RedSafe^™^ Nucleic Acid Staining Solution (iNtRON Biotechnology, Gyeonggi, South Korea). This diagnostic technique has been previously used in numerous studies to screen *Francisella* spp., including *F. tularensis* [24,25,26,27].

## 3. Results and Discussion

None of the 774 (0.0%; 95%CI: 0.0–0.5) wild rabbits and 178 (0.0%; 95%CI: 0.0–2.1) Iberian hares analyzed showed *F. tularensis* DNA. Although the complementary use of qPCR could have strengthened the data we present herein, our results indicate the absence or very low circulation of the bacterium in wild lagomorph populations in the study area.

Of the 952 lagomorphs sampled, excluding missing values, 683 (68.3%) were adults, 231 (24.7%) were subadults, and 65 (7.0%) were juveniles. Furthermore, the sampling was balanced in terms of sex, with 362 (47.0%) males and 409 (53.0%) females among the European wild rabbits. As for the Iberian hares, 89 (53.0%) were males, and 79 (47.0%) were females.

To the best of the authors’ knowledge, this is the first large-scale study assessing *F. tularensis* circulation in wild lagomorphs in Andalusia. In this Iberian region, the European wild rabbit and the Iberian hare are major small game species, being a significant food source for humans, often consumed without veterinary inspection [28]. These wild lagomorph species can act as reservoirs for vector-borne zoonotic pathogens [29,30,31]. In this regard, both the European wild rabbit and the Iberian hare have been suggested to play a role in the sylvatic cycle of several zoonotic pathogens in southern Spain, such as *Coxiella burnetii*, *Sarcoptes scabiei*, and *Leishmania infantum* [32,33,34]. Despite not finding *F. tularensis* DNA in the present study, endemic distribution of *F. tularensis* was found in these wild lagomorph species in northwest Spain, with prevalences of 1.4% (21/1492) for European wild rabbits and 9.6% (135/1403) for Iberian hares [17]. These findings point to differences in the geographical distribution of the bacterium across the Iberian Peninsula. Similarly, a heterogeneous distribution of *F. tularensis* was found in European brown hare (*Lepus europaeus*) in France [35]. Differences in the spatial distribution of *F. tularensis* might be due to the abundance of natural reservoir species or environmental factors enhancing the transmission of the bacterium in specific epidemiological scenarios [13]. In this regard, it has been described that large tularemia outbreaks in humans in northwestern Spain presented an association with common vole (*Microtus arvalis*) abundance, which favors disease transmission and spillover contamination in the environment, likely affecting Iberian hare populations of that region [36,37,38]. In this sense, *F. tularensis* has a high survival capacity in humid and cold environments, with water being a key factor in the spread of the disease. The study region is mainly characterized by dry, warm summers and mild winters, which do not favor the bacterium’s survival in the environment [18].

The results obtained in the present study, together with those observed by Minguez-González et al. [17] in northwest Spain, denote spatial variations in the risk of zoonotic transmission of *F. tularensis* from wild lagomorphs throughout the Iberian Peninsula. Of note, these findings align with the geographical distribution of human tularemia cases in Spain, where most outbreaks have been recorded in the northwest, with no cases being reported in southern Spain so far [16]. Human outbreaks are often preceded by the appearance of the disease in animals [8]. Thus, it has already been noted that the multiannual cyclicity of vole outbreaks may serve as a basis for predicting the risk of human tularemia in Spain [36]. In this sense, the European brown hare has been shown to be a useful sentinel species for *F. tularensis* surveillance in France [13]. However, while this species is found only in the north of Spain, the Iberian hare is present in almost the entire Iberian Peninsula, and the European wild rabbit is the most abundant lagomorph species in this European region. Further epidemiological studies are warranted to assess the circulation of this bacterium in wild lagomorph species and to estimate the risk of exposure for humans from these species in the Iberian Peninsula [7].

## 4. Conclusions

In summary, our findings indicate the absence or very low circulation of *F. tularensis* in European wild rabbit and Iberian hare populations, denoting a limited risk of *F. tularensis* transmission from these lagomorphs to other sympatric species, including humans, in the Mediterranean ecosystems of southern Spain. However, given the potential usefulness of wild lagomorphs as sentinel species of this zoonotic bacterium, large-scale monitoring programs should be implemented and maintained in wild lagomorph populations in the Iberian Peninsula.

## Figures and Tables

**Figure 1 animals-14-03376-f001:**
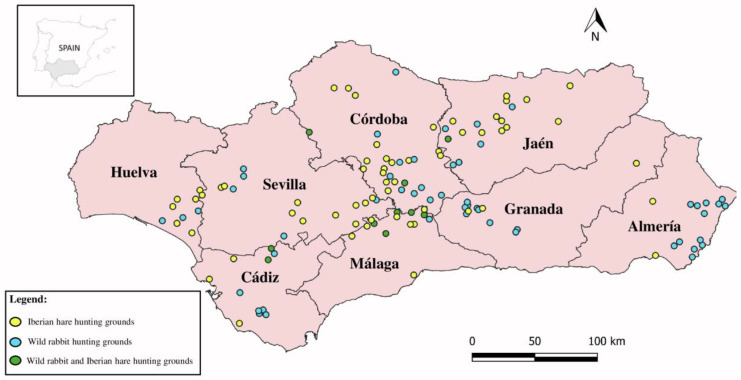
Spatial distribution of sampled hunting grounds in the study region (Andalusia, southern Spain).

## Data Availability

The data presented in this study are available on request from the corresponding author. The data are not publicly available due to privacy restrictions and the long extension of datasets.

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
