# Peer review of "Francisella tularensis in Wild Lagomorphs in Southern Spain’s Mediterranean Ecosystems"

_animals, 2024, doi:10.3390/ani14233376_

Round 1
Reviewer 1 Report
Comments and Suggestions for Authors
The manuscript (MS) entitled "Francisella tularensis in wild lagomorphs in southern Spanish Mediterranean ecosystems" described by authors Sabrina Castro-Scholten et al. indicated low to no circulation of F. tularensis in European wild rabbit and Iberian hare populations in southern Spain. They tried to detect F. tularensis DNA from 774 European wild rabbits and 178 Iberian hare, which were collected during hunting season between 2017 and 2023, by use of conventional PCR targeting 16S rRNA gene. The results and discussion is clear but I have some questions.
Major comments.
About study area and control. Tularemia occurrences in Spain were reported mainly northern part and no human case of tularemia in southern Spain, the study area of this report. Authors would be better to test samples from animals captured in northern or middle areas of Spain or use DNA purified from F. tularensis infected animal as control.
About experiment. In this decade, I have never seen the epidemiological survey of tularemia using conventional-PCR only. Authors should perform real-time PCR as well. Otherwise, detection limit of the PCR should be shown by use of positive control DNA, DNA purified from spleen spiked with F. tularensis DNA or Spanish isolate. Because this is negative result, detection limit of the PCR, quality control should be managed precisely. Amount of DNA applied on PCR assays for each sample should be described as well.
About collaborators and biosafty. I speculate that the animals might be shot by hunters. However, there are no comments in acknowledgments to hunter or hunters association. If any hunters were cooperated, acknowledgments should be added. In that case, I wonder if the authors encouraged hunters to wear groves, mask, and eyewear when dissecting animals and how legally transport the sample to laboratory. If sampling were performed by surveillance program members for the purpose of surveillance only, permission No. of animal experiment should be described.
Minor comments
It is interesting that animals tested were apparently healthy or abnormal before captured. Generally, it is extremely rare to detect F. tularensis from healthy animals by conventional-PCR but comparatively easy to detect from dead or unwell animals. I think it is cost-ineffective to survey F. tularensis DNA from normal live animals. Serological survey will be first choice.
Author Response
Answer to Reviewer’s 1 Comments:
Comment 1: The manuscript (MS) entitled "Francisella tularensis in wild lagomorphs in southern Spanish Mediterranean ecosystems" described by authors Sabrina Castro-Scholten et al. indicated low to no circulation of F. tularensis in European wild rabbit and Iberian hare populations in southern Spain. They tried to detect F. tularensis DNA from 774 European wild rabbits and 178 Iberian hare, which were collected during hunting season between 2017 and 2023, by use of conventional PCR targeting 16S rRNA gene. The results and discussion is clear but I have some questions.
Response 1: We would like to thank the reviewer for their positive feedback and for the comments on how the manuscript can be improved.
Major comments:
Comment 2: About study area and control. Tularemia occurrences in Spain were reported mainly northern part and no human case of tularemia in southern Spain, the study area of this report. Authors would be better to test samples from animals captured in northern or middle areas of Spain or use DNA purified from F. tularensis infected animal as control.
Response 2: We agree with the reviewer that several studies conducted in northern Spain described F. tularensis endemic circulation in different species, including wild lagomorphs (Mínguez-González et al., 2021). In view of the importance that wild lagomorphs present in southern Spain, due to their abundance and hunting significance in this region, the aim of this study was to assess the circulation of this bacterium specifically in lagomorph populations in southern Spain, where no epidemiological surveys of F. tularensis have been conducted so far.
The positive control was an infected Hyalomma lusitanicum tick collected in Spain, already published in Díaz-Sánchez et al., 2021 (GenBank Accession Number: MT386092). Additional information has been added to clarify this point (lines 124-126).
Comment 3: About experiment. In this decade, I have never seen the epidemiological survey of tularemia using conventional-PCR only. Authors should perform real-time PCR as well.
Response 3: In the present study, we have screened F. tularensis in wild lagomorphs using a broad-spectrum PCR which targets the 16S ribosomal gene of the Francisella genus. We disagree with the reviewer since this PCR has been previously used in numerous studies to screen Francisella spp, including F. tularensis, during the last few years (Lefcort et al., 2020; Chang et al., 2021; Usananan et al., 2022; Schulze et al., 2016). Indeed, Schulze et al. (2016) used only the conventional PCR as screening tool for F. tularensis and they used qPCR only to confirm the presence of F. tularensis DNA in suspected positive samples. Additional information has been included in the material and method section (lines 131-133).
Considering the existing scientific literature and given that evaluating the sensitivity of the PCR used is outside the scope of the present survey, we believe that our diagnostic technique is suitable for its intended purpose. However, at the Editor's discretion, we could conduct additional analyses using a qPCR protocol, though this will require extra time in the review process.
Consequently, we have included an additional sentence in the discussion section pointing that an additional qPCR might have strengthened the data we present (lines 136-138).
Comment 4: Otherwise, detection limit of the PCR should be shown by use of positive control DNA, DNA purified from spleen spiked with F. tularensis DNA or Spanish isolate. Because this is negative result, detection limit of the PCR, quality control should be managed precisely.
Response 4: We agree with the reviewer. However, the calculation of the limit of detection has not been possible to carry out due to the reasons mentioned in the previous response. This calculation could only be performed with quantitative PCR protocols and with the knowledge of absolute quantification of the target DNA to create the appropriate calibration curve and serial dilution.
Comment 5: Amount of DNA applied on PCR assays for each sample should be described as well.
Response 5: Now, in the text we have specified the amount of template added to the PCR reactions (in this case 8 uL).
Comment 6: About collaborators and biosafty. I speculate that the animals might be shot by hunters. However, there are no comments in acknowledgments to hunter or hunters association. If any hunters were cooperated, acknowledgments should be added. In that case, I wonder if the authors encouraged hunters to wear groves, mask, and eyewear when dissecting animals and how legally transport the sample to laboratory. If sampling were performed by surveillance program members for the purpose of surveillance only, permission No. of animal experiment should be described.
Response 6: It is correct that animals are shot by hunters and we thank to the reviewer for reminding as to include them in the acknowledgements section. We would like to point out that although the animals were shot by hunters, necropsy and sampling were carried out by members of our research group. After that, carcasses were returned to the hunter, so it was not necessary to indicate biosafety measures, beyond those recommended by the Spanish Ministry of Agriculture, Fisheries and Food (MAPA), and samples were kept refrigerated until arrival at the laboratory, where they were frozen at -80ºC until molecular analysis were undertaken. In line with this, the animals sampled were legally harvested during the hunting seasons and no animal was only hunted for surveillance purposes, therefore a permission number was not necessary. Additional information has been included to clarify this point (lines 102-103).
Minor comments:
Comment 7: It is interesting that animals tested were apparently healthy or abnormal before captured. Generally, it is extremely rare to detect F. tularensis from healthy animals by conventional-PCR but comparatively easy to detect from dead or unwell animals. I think it is cost-ineffective to survey F. tularensis DNA from normal live animals. Serological survey will be first choice.
Response 7: We agree with the reviewer that it is easier to detect F. tularensis circulation in animals found dead (i.e. passive surveillance). However, this study was designed based on active surveillance, which allowed structured, purposeful data collection, in contrast to the often irregular and incomplete nature of passive surveillance. In fact, active surveillance allows the search for pathogens in order to obtain data on their prevalence, risk factors and geographical distribution. We do not agree that it is ineffective to survey F. tularensis from normal live animals, since previous studies carried out on healthy hunted lagomorphs in Spain (Mínguez-González et al., 2021) and Portugal (Lopes et al., 2016) detected F. tularensis DNA.
On the other hand, we also agree that serosurveys could present advantages over molecular studies in determining the exposure of a population to a pathogen. Nevertheless, in the present study, we considered that a molecular survey was the best choice because molecular assays do not depend on the species analyzed, unlike serological tests such as ELISA, which are species specific and are not available and validated for wild lagomorphs. In fact, the most used serological tests to detect anti-F. tularensis antibodies are agglutination, which could present low specificity when sera with poor quality is used, as is common in samples from wildlife collected post-mortem.
Reviewer 2 Report
Comments and Suggestions for Authors
This work analyses a considerable number of animals of two species of lagomorphs over six years in southern Spain, to detect the presence of Francisella tularensis. The authors indicate that they calculated a sample size that ensured they could detect at least one infected animal. In addition, the animals came from a wide area within the study region. However, since there is spatial and temporal heterogeneity related to the origin of sampled animals, the authors make a risky claim when they affirm that there is no circulation of the bacteria in the study area.
Since there is no explanation about how the spleen sample is handled between the time of dead of the animal and the time the organ is collected and frozen, I wonder is there is risk that the biological material has deteriorated, preventing that samples with low bacterial loads being detected. I suggest to the authors to indicate the details of this procedure, and if there is variability between the time of death and organ collection, to mention it as a potential source of bias. In addition, it would be appropriate to indicate whether an internal amplification control was used to verify that there were no samples with PCR inhibitors.
The authors mention in the discussion that the difference detected in the frequency of infection in lagomorphs from the south and north of Spain may be due to differences in the presence of vectors or other natural reservoirs of the bacteria, and environmental variables. However, of these three elements, they only explain characteristics of the reservoirs of the bacteria described in this area of ​​Europe, but they do not mention what is known about the presence of vectors or potential vectors in the south of Spain, nor what environmental characteristics are relevant in explaining this low circulation of the pathogen. The authors should discuss these two aspects to develop more specific explanations that guide future studies. Besides, I suggest that the authors rephrase the affirmation about the absence of circulation of the bacteria in Southern Spain, taking into account the limitations of the sampling strategy and the heterogeneity in the origin of sampled animals.
In the introduction and/or discussion segment, it should be provides more details to understand the wild transmission cycle of F. tularensis, including the characteristics of relevant natural reservoirs and vectors, as well as the relevance of the environmental component.
Table 1 has no title and considering that the presence of the bacteria was not detected, it does not contribute to the work. However, some of the data in this table could be explained in the results (for example, distribution by age or sex).
The results and discussion segment should start with the most relevant information that resulted from the study: it was not detected DNA of F. tularensis in the analyzed samples. From that point, the other elements mentioned in the text can be discussed.
In the sample collection and processing section, more details should be given regarding the timing between the death of the animals and the collection/freezing of the samples.
In the molecular detection section, more details should be provided regarding the PCR protocol used, such as primer concentration and volume of template used in each reaction. In addition, if the concentration and quality of the extracted DNA was measured, this should also be indicated.
Author Response
Answer to Reviewer’s 2 Comments:
Comment 1: This work analyses a considerable number of animals of two species of lagomorphs over six years in southern Spain, to detect the presence of Francisella tularensis. The authors indicate that they calculated a sample size that ensured they could detect at least one infected animal. In addition, the animals came from a wide area within the study region. However, since there is spatial and temporal heterogeneity related to the origin of sampled animals, the authors make a risky claim when they affirm that there is no circulation of the bacteria in the study area.
Response 1: We would like to thank the reviewer for the comments on how the paper could be improved.
The heterogeneous distribution of samples is a result of the uneven distribution of wild lagomorph populations across the study region, especially the Iberian hare, which suffered a drop in its populations or disappeared from certain areas of Andalusia after the outbreak of myxomatosis virus that began in this region in 2018 (García-Bocanegra et al., 2019). Nevertheless, following the reviewer’s suggestion we also indicate that the circulation could be very low in this wild lagomorph populations (lines 137-138).
Comment 2: Since there is no explanation about how the spleen sample is handled between the time of dead of the animal and the time the organ is collected and frozen, I wonder is there is risk that the biological material has deteriorated, preventing that samples with low bacterial loads being detected. I suggest to the authors to indicate the details of this procedure, and if there is variability between the time of death and organ collection, to mention it as a potential source of bias. In addition, it would be appropriate to indicate whether an internal amplification control was used to verify that there were no samples with PCR inhibitors.
Response 2: Although there is a time gap between sample collection and storage, this was as brief as possible and never higher than four hours. During this time, spleen samples are kept refrigerated in transportable coolers with frozen cold accumulators inside. Therefore, we consider that the risk of sample deterioration is low. Additional information has been added to clarify this point (lines 102-103).
Although we did not use internal controls, we measured concentration and quality of all DNA elutions using the NanoDrop ND-2000 spectrophotometer (Thermo Fisher Scientific, Waltham, MA). Values of 260/280 of all extractions were within the expected range and the DNA presented a mean concentration of 407.7 ng/µl. This information is shown in Supplementary Table S1. Additionally, the extraction kits used in the present study specifically reduce the presence of PCR inhibitors and the extracted DNA has been used in previous studies for the detection of Anaplasma bovis (Remesar et al., 2024) Coxiella burnetii (Castro-Scholten et al., 2024) and Leishmania infantum (Barbero-Moyano et al., 2024) in which prevalences of 9.4%, 17.9% and 59.0% were detected, respectively. For all these reasons, we consider that negative samples were not related to the potential presence of PCR inhibitors.
Comment 3: The authors mention in the discussion that the difference detected in the frequency of infection in lagomorphs from the south and north of Spain may be due to differences in the presence of vectors or other natural reservoirs of the bacteria, and environmental variables. However, of these three elements, they only explain characteristics of the reservoirs of the bacteria described in this area of ​​Europe, but they do not mention what is known about the presence of vectors or potential vectors in the south of Spain, nor what environmental characteristics are relevant in explaining this low circulation of the pathogen. The authors should discuss these two aspects to develop more specific explanations that guide future studies. Besides, I suggest that the authors rephrase the affirmation about the absence of circulation of the bacteria in Southern Spain, taking into account the limitations of the sampling strategy and the heterogeneity in the origin of sampled animals.
Response 3: Based on the reviewer’s suggestion, information on environmental characteristics relevant in the circulation of this pathogen have been added to the result and discussion section (lines 163-167). However, due to the presence of competent vectors in both the study region and northern Spain, this has been ruled out as a possible cause of the heterogeneous distribution of F. tularensis.
In addition, we have rephrased the affirmation about the absence of circulation to: “which indicates absence or very low circulation of the bacterium in wild lagomorph populations in the study area” (lines 137-138).
Comment 4: In the introduction and/or discussion segment, it should be provides more details to understand the wild transmission cycle of F. tularensis, including the characteristics of relevant natural reservoirs and vectors, as well as the relevance of the environmental component.
Response 4: In accordance with the reviewer’s suggestion additional information related to the cycle of F. tularensis has been added to the introduction section (lines 55-63).
Comment 5: Table 1 has no title and considering that the presence of the bacteria was not detected, it does not contribute to the work. However, some of the data in this table could be explained in the results (for example, distribution by age or sex).
Response 5: As recommended by the reviewer the table has been removed and additional data has been included in the result and discussion section (lines 139-143).
Comment 6: The results and discussion segment should start with the most relevant information that resulted from the study: it was not detected DNA of F. tularensis in the analyzed samples. From that point, the other elements mentioned in the text can be discussed.
Response 6: Following the reviewer’s suggestion, the beginning of the results and discussion section has been modified to: “None of the 774 (0.0%; 95%CI: 0.0-0.5) wild rabbits and 178 (0.0%; 95%CI: 0.0-2.1) Iberian hares analyzed showed F. tularensis DNA, which indicates absence or very low circulation of the bacterium in wild lagomorph populations in the study area”.
Comment 7: In the sample collection and processing section, more details should be given regarding the timing between the death of the animals and the collection/freezing of the samples.
Response 7: The timing between the death of the animals and necropsy and sample collection can be considered short, at most 1 hour, since the personnel who performed the sampling moved to the hunting grounds and the hunters handed over the animals after hunting them. Following this, the time gap between sample collection and storage was as brief as possible and never higher than four hours. During this time, spleen samples are kept refrigerated in transportable coolers with frozen cold accumulators inside. Additional information has been added to clarify this point (lines 102-103).
Comment 8: In the molecular detection section, more details should be provided regarding the PCR protocol used, such as primer concentration and volume of template used in each reaction. In addition, if the concentration and quality of the extracted DNA was measured, this should also be indicated.
Response 8: Following the reviewer’s suggestion more details have been provided regarding the PCR protocol used and the concentration and quality of the extracted DNA (lines 115-126).
Round 2
Reviewer 1 Report
Comments and Suggestions for Authors
Thank you for your kind reply and improvement.
Now, my questions were cleared up and I understood your research.
I have no compliant and agree to publish the manuscript on MDPI animal.